# Multifaceted membrane interactions of human Atg3 promote LC3-phosphatidylethanolamine conjugation during autophagy

Yansheng Ye[1,5], Erin R. Tyndall[1,5], Van Bui[2,5], Maria C. Bewley[1], Guifang Wang[1], Xupeng Hong[3], Yang Shen[4], John M. Flanagan[1], Hong-Gang Wang [2] ✉ & Fang Tian [1] ✉

Autophagosome formation, a crucial step in macroautophagy (autophagy), requires the covalent conjugation of LC3 proteins to the amino headgroup of phosphatidylethanolamine (PE) lipids. Atg3, an E2-like enzyme, catalyzes the transfer of LC3 from LC3-Atg3 to PEs in targeted membranes. Here we show that the catalytically important C-terminal regions of human Atg3 (hAtg3) are conformationally dynamic and directly interact with the membrane, in collaboration with its N-terminal membrane curvature-sensitive helix. The functional relevance of these interactions was confirmed by in vitro conjugation and in vivo cellular assays. Therefore, highly curved phagophoric rims not only serve as a geometric cue for hAtg3 recruitment, but also their interaction with hAtg3 promotes LC3-PE conjugation by targeting its catalytic center to the membrane surface and bringing substrates into proximity. Our studies advance the notion that autophagosome biogenesis is directly guided by the spatial interactions of Atg3 with highly curved phagophoric rims.

Macroautophagy (autophagy) is a major intracellular degradation and recycling process initiated by forming a cup-like membrane structure known as the phagophore. As the phagophore expands, cytosolic components that are targeted for degradation, such as aggregated proteins, damaged organelles, and foreign organisms including viruses and bacteria, are engulfed. Following the phagophore elongation, it is sealed to create a double-membrane vesicle called the autophagosome. The autophagosome is then delivered to the lysosome to degrade and recycle its contents[1-5]. In eukaryotes, autophagy is essential for maintaining cellular homeostasis and basic functions; its dysfunction leads and/or contributes to many disease processes including neurodegeneration, infection, cardiovascular impairment, and tumorigenesis[6-10].

A key step in autophagosome biogenesis is the conjugation of LC3 family proteins to phosphatidylethanolamine (PE)[5,11]. This LC3-PE conjugate regulates phagophore expansion and serves as a docking site for autophagic cargos[12-14]. This conjugate reaction is catalyzed by three ubiquitin-like conjugation enzymes: the E1-like Atg7, the E2-like Atg3, and the E3-like Atg12-Atg5/Atg16 complex. Despite their conceptual similarities, Atg7, Atg3, and Atg12-Atg5/Atg16 are structurally and functionally distinct from canonical E1, E2, and E3 enzymes[15-22]. For instance, unlike canonical E2 enzymes, in vitro Atg3 alone selects substrate PE lipids and can catalyze the transfer of LC3 from an LC3-Atg3 intermediate to PEs without the Atg12-Atg5/Atg16 complex[23-25]. As this reaction occurs at the membrane surface with PEs in a lipid

[1]Department of Biochemistry and Molecular Biology, Pennsylvania State University College of Medicine, Hershey, PA, USA. [2]Department of Pediatrics, Division of Pediatric Hematology and Oncology, Pennsylvania State University College of Medicine, Hershey, PA, USA. [3]Department of Microbiology and Immunology, Pennsylvania State University College of Medicine, Hershey, PA, USA. [4]Laboratory of Chemical Physics, National Institute of Diabetes and Digestive and Kidney Diseases, US National Institutes of Health, Bethesda, MD, USA. [5]These authors contributed equally: Yansheng Ye, Erin R. Tyndall, Van Bui. ✉e-mail: huw11@psu.edu; ftian@psu.edu

bilayer phase, the protein-lipid interaction of Atg3 plays an essential role in its function. In fact, Atg3's function requires the presence of an N-terminal amphipathic helix (NAH) that selectively interacts with membranes that contain lipid-packing defects, such as the leading edge of the growing phagophore where membranes are highly curved[26–29]. Our recent studies have further revealed that the membrane association of the NAH of human Atg3 (hAtg3) is necessary but insufficient to catalyze LC3-PE conjugation[30]. We have discovered a conserved region following the hAtg3 NAH that tightly coordinates its membrane geometry-sensitive interaction with its catalytic activity. In addition, enhancing the interaction of Atg3 with membranes by acetylation of its NAH reportedly extends the duration of autophagy[31]. However, how the C-terminal located catalytic center is directed to the membrane surface for effective LC3-PE conjugation remains obscure.

In this study, we have used NMR spectroscopy to determine the solution structure of hAtg3 and discovered that both its NAH and catalytically important C-terminal regions directly interact with the membrane. The functional significance of this interaction was confirmed by in vitro and in vivo loss-of-function mutational analyses. Our results suggest that hAtg3 exploits a multifaceted membrane-association mechanism[32–38] to position the catalytic residue Cys264 at the membrane surface and bring substrates, LC3 and PE lipids, into proximity to promote LC3-PE conjugation. In addition, NMR hydrogen/deuterium exchange studies revealed that the catalytic loop and following α-helix of hAtg3 are conformationally dynamic in aqueous solution and are conducive to interaction with the membrane, even though the general hAtg3 core structure is similar to those previously reported for yeast and *Arabidopsis thaliana*[18,39]. Together, our studies advance an emerging concept that the interactions of Atg3 with the highly curved membrane rims of the phagophore spatially regulate autophagosome biogenesis[29,40].

## Results

### Structure and conformational plasticity of hAtg3

hAtg3 is designated as an intrinsically disordered protein since more than 1/3 of its 314 residues are in unstructured regions[41]. In an earlier work, we showed that residues 90 to 190 form an unstructured loop and that deletion of these amino acids (hAtg3$^{\Delta 90-190}$) minimally perturbed hAtg3's core structure and did not affect its function in vitro (Supplementary Figs. 6–8[30]). We have also shown that residues 1 to 25 are unstructured in aqueous solution but undergo a conformational change to form an amphipathic α-helix upon interaction with the membrane[42]. Deletion of both regions (hAtg3$^{\Delta N25,\ \Delta 90-190}$) causes a minimal perturbation in the core structure as evidenced by an overlay of the TROSY spectra of hAtg3$^{\Delta 90-190}$ and hAtg3$^{\Delta N25,\ \Delta 90-190}$ shown in Supplementary Fig. 1. Since the hAtg3$^{\Delta N25,\ \Delta 90-190}$ construct produced high-resolution spectra, it was used for NMR structure determination. This construct is very similar to a yeast Atg3 (yAtg3) version that was recently selected for X-ray structure determination[43].

An overlay of ten solution structures of hAtg3$^{\Delta N25,\ \Delta 90-190}$ (determined with 2360 NOEs, 264 torsion angle restraints, and 552 RDCs measured in four alignment media) is shown in Fig. 1a; statistics from our NMR structure determination are summarized in Supplementary Table 1. Superpositions of hAtg3$^{\Delta N25,\ \Delta 90-190}$ onto the structures of yeast and *Arabidopsis thaliana* Atg3 (yAtg3 and AtAtg3) demonstrate a conserved E2-like fold. The reported Cα RMSDs of hAtg3 to yAtg3 and AtAtg3 structures by Chimera are 1.1 Å and 1.0 Å, respectively. The most notable deviations are observed in helix F (according to the secondary structure definition in the yAtg3[18]), which immediately follows the catalytic loop containing Cys264 (Fig. 1b). In addition, in the structure of hAtg3 determined here, helix F (residues 268–278) is notably less well defined with an RMSD of 0.9 Å while the rest of the structured elements converge to an overall RMSD of 0.4 Å. This is due to the use of a limited number of observable NOEs for the structure calculation and may reflect the flexibility of this helix.

To characterize the dynamic features of hAtg3$^{\Delta N25,\ \Delta 90-190}$, we performed NMR hydrogen/deuterium (H/D) exchange experiments. Instead of quantitatively determining an exchange rate constant, for the purpose of this discussion, we classified the peaks observed in TROSY spectra into three groups according to their intensities in H/D exchange experiments. Group I amides include those that are in fast exchange (colored red in Fig. 1c) and have lost >67% of their peak intensities in the first 2D TROSY spectrum (data collection started ~21 min after the addition of a D$_2$O buffer and took ~17 min to finish) relative to a reference spectrum collected in an H$_2$O buffer. Group II amides exchanged slowly (colored cyan in Fig. 1c); more than 33% of their initial peak intensities remained in the last 2D TROSY spectrum taken after ~23 h of H/D exchange. Group III amides displayed intermediate H/D exchanges, slower than Group I and faster than Group II (colored orange in Fig. 1c). Most Group I amides are in unstructured loops, but surprisingly, the amides of residues in helix F also exhibit fast exchange, suggesting that this region is dynamic. Group II amides are generally located in β-sheets and helices B (residues 36–49) and G (residues 289–297). Group III amides are distributed throughout the structure. These results indicate that helix F is more flexible than the other two long helices, B and G. In addition, as we reported previously, the catalytic loop (residues 262–267) is likely involved in motions on the millisecond timescale since its NH peaks were not observed in the TROSY spectrum due to exchange broadening[30]. Taken together, our NMR results provide direct evidence that the catalytic loop and adjacent regions of hAtg3 are highly dynamic in aqueous solution. As we describe below, the structural changes in these regions are observed upon direct interaction with the membrane.

### Multifaceted membrane association of hAtg3

The LC3-PE conjugation reaction requires that PEs are anchored in lipid bilayers. However, the mechanism by which the catalytic site Cys264 of hAtg3 comes into proximity with the bilayer surface to facilitate the transfer of LC3 to PEs remains unknown. We have previously reported that there are substantial chemical shift perturbations (CSPs) around the catalytic site of hAtg3 upon the binding of its NAH to bicelles (as we demonstrated previously, bicelles support the conjugase activity of hAtg3)[30]. For these experiments we used the hAtg3$^{\Delta 90-190}$ construct because the membrane interaction requires residues 1 to 25. Despite extensive effort, we could assign only 155 out of 199 non-proline residues of the bicelle-bound hAtg3$^{\Delta 90-190}$; 22 out of 38 residues from Ser260 to Val297 (including residues in the catalytic loop, helices F and G) remain unassigned[42]. Resonances from these residues are either missing or display pronounced exchange broadening in the TROSY spectrum. Thus, these regions are likely involved in conformational exchanges in bicelle-bound hAtg3$^{\Delta 90-190}$. During sample optimization, we noticed that hAtg3$^{\Delta 90-190}$ in bicelles is stable only for one or two days at pH 6.5, but at pH 7.5, it remains stable for several weeks. We hypothesized that the membrane interaction of a protonated His residue in hAtg3$^{\Delta 90-190}$ contributes to its instability at lower pH. This observation led us to substitute His266, one of two fully conserved His residues (Supplementary Fig. 2), to Leu to mimic its non-protonated state. Like the wildtype protein, hAtg3$^{H266L}$ (but not hAtg3$^{H266K}$) shows similar conjugase activity in an in vitro LC3B-PE conjugation assay (Supplementary Fig. 3a, b). Moreover, the hAtg3$^{\Delta 90-190,\ H266L}$ construct shows dramatically improved spectral quality in bicelles. Multiple new resonances were observed and assigned to the hAtg3 catalytic region (Supplementary Fig. 3c). Further optimizations led to an hAtg3 construct containing four mutations (H240Y, V241A, P263G, and H266L, referred to as hAtg3$^{\Delta 90-190,\ 4M}$) that produced a high-quality TROSY spectrum (Supplementary Fig. 4a) and its corresponding hAtg3$^{4M}$ construct remained functional (Supplementary Fig. 3a, b). We have assigned all but 13 residues of the bicelle-bound hAtg3$^{\Delta 90-190,\ 4M}$ to date. Unassigned residues are Met1, Gln2, Asn3, Lys27, Phe28,

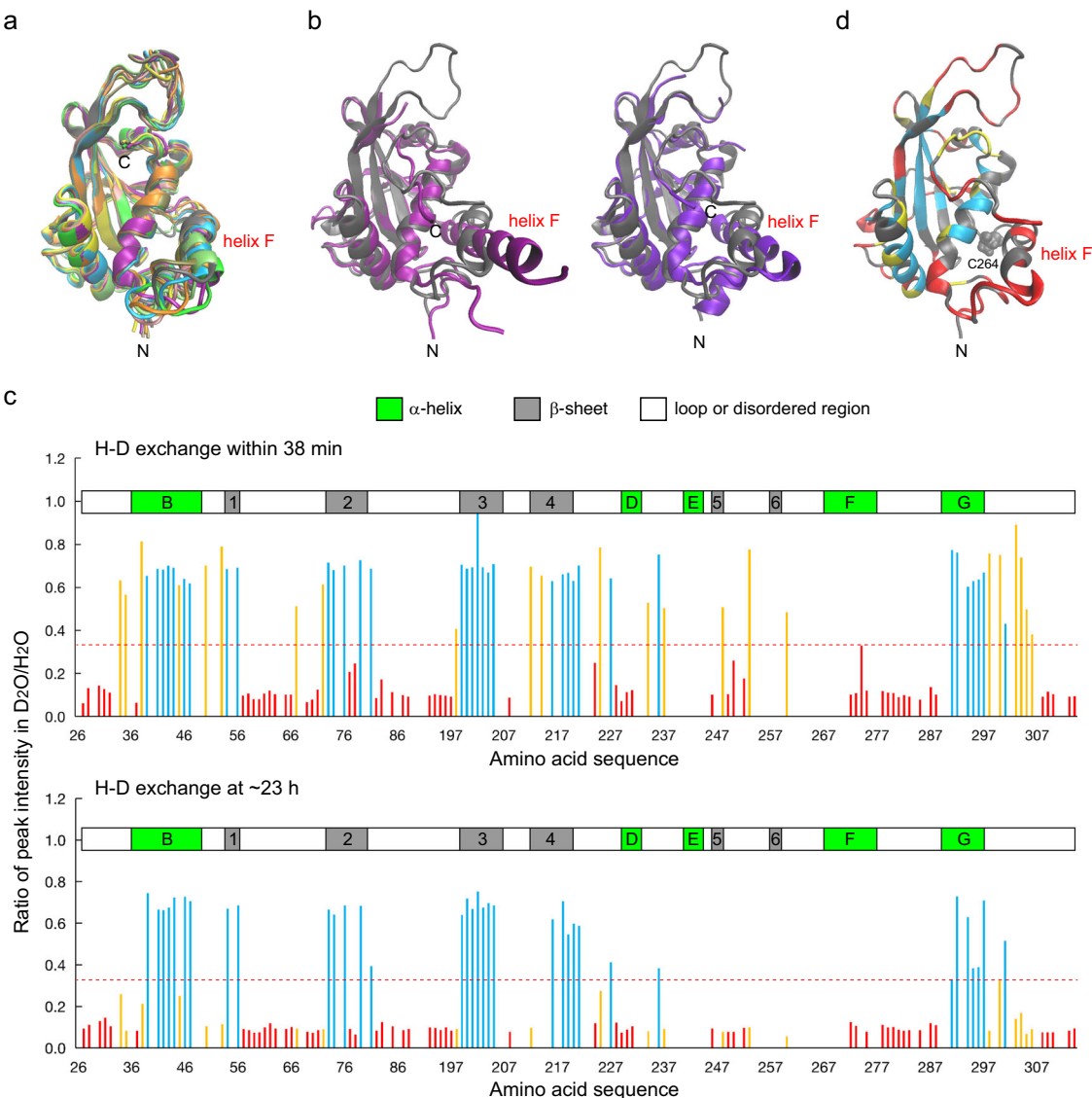

**Fig. 1 | Structure and conformational flexibility of hAtg3^{ΔN25, Δ90–190}. a** Overlay of hAtg3^{ΔN25, Δ90–190} ten solution structures. C-terminal unstructured residues 306–314 are not shown for clarity. **b** Structural alignments of hAtg3^{ΔN25, Δ90–190} (gray) with yeast Atg3 (purple, PDB 6OJJ, left), and *Arabidopsis thaliana* Atg3 (violet, PDB 3VX8, right). Helix F is indicated. **c** H/D exchange of ^{15}N-labeled hAtg3^{ΔN25, Δ90–190} at pH 6.5, 25 °C. Group I: residues with fast H/D exchange (more than 67% of resonance intensities lost within 38 min after adding D$_2$O) are colored red. Group II: residues with slow H/D exchange (less than 67% of resonance intensities lost at ~23 h after adding D$_2$O) are colored cyan. Group III: residues with intermediate H/D exchange (slower than Group I and faster than Group II) are colored orange. Given the structural conservation, secondary structural elements are indicated according to corresponding elements in a yeast Atg3 structure (PDB 2DYT). Source data are provided as a Source Data file. **d** Residues with different H/D exchange rates are mapped onto the hAtg3^{ΔN25, Δ90–190} NMR structure in corresponding colors as shown in (**c**). Residues with uncharacterized H/D exchange rates are shown in gray. Helix F is indicated, and catalytic residue Cys264 is shown in VDW.

Ala65, Asp239, Lys242, Lys243, Cys264, Arg265, His311, and Phe312. Importantly, most residues adjacent to the catalytic region, including those in helices F and G, are now assigned; this allows us to analyze their perturbations upon interacting with the membrane. Supplementary Fig. 4b plots the CSPs of hAtg3^{Δ90–190, 4M} induced by bicelles. Consistent with our previous study of wildtype protein[30], residues showing substantial CSPs include both NAH and C-terminal residues. This observation suggests that they may participate in the membrane-interacting surface.

To determine the residues in hAtg3 that directly interact with the membrane, we performed an NMR cross-saturation experiment. This experiment was initially developed to map protein-protein interfaces and has been used to identify surface residues that are directly involved in protein/lipid interactions[44–47]. On a sample of perdeuterated ^{15}N, ^2H- hAtg3^{Δ90–190, 4M} mixed with 12% (w/v) bicelles (DMPC:DMPG:DHPC = 8:2:20, molar ratio, $q = 0.5$) in an 80% D$_2$O and

20% H$_2$O buffer, irradiation at 1.28 ppm (hydrophobic tail residues of lipids) with a 40 ms Gaussian pulse for 1.6 s resulted in the effective saturations of all lipid resonances, except the peak from choline methyl groups, due to spin diffusion (Supplementary Fig. 5a). The selective transfer from the saturated lipid resonances to the residues in hAtg3 located at the protein/lipid interface results in a reduction of their resonance intensities when compared to spectra with off-resonance saturations. Figure 2a, b shows the results of these cross-saturation experiments for hAtg3^{Δ90–190, 4M}. As expected, residues in the NAH display the largest cross-saturation effects since they form an amphipathic helix that is inserted into the membrane. Surprisingly, two stretches of C-terminal residues also display significantly reduced intensities. Region I includes residues from 262 to 277 which are in the catalytic loop and dynamic helix F (Fig. 1c). Region II spans residues from 291 to 300; most of these reside in helix G (Fig. 1c). When mapped onto the hAtg3 solution structure, these residues appear to cluster

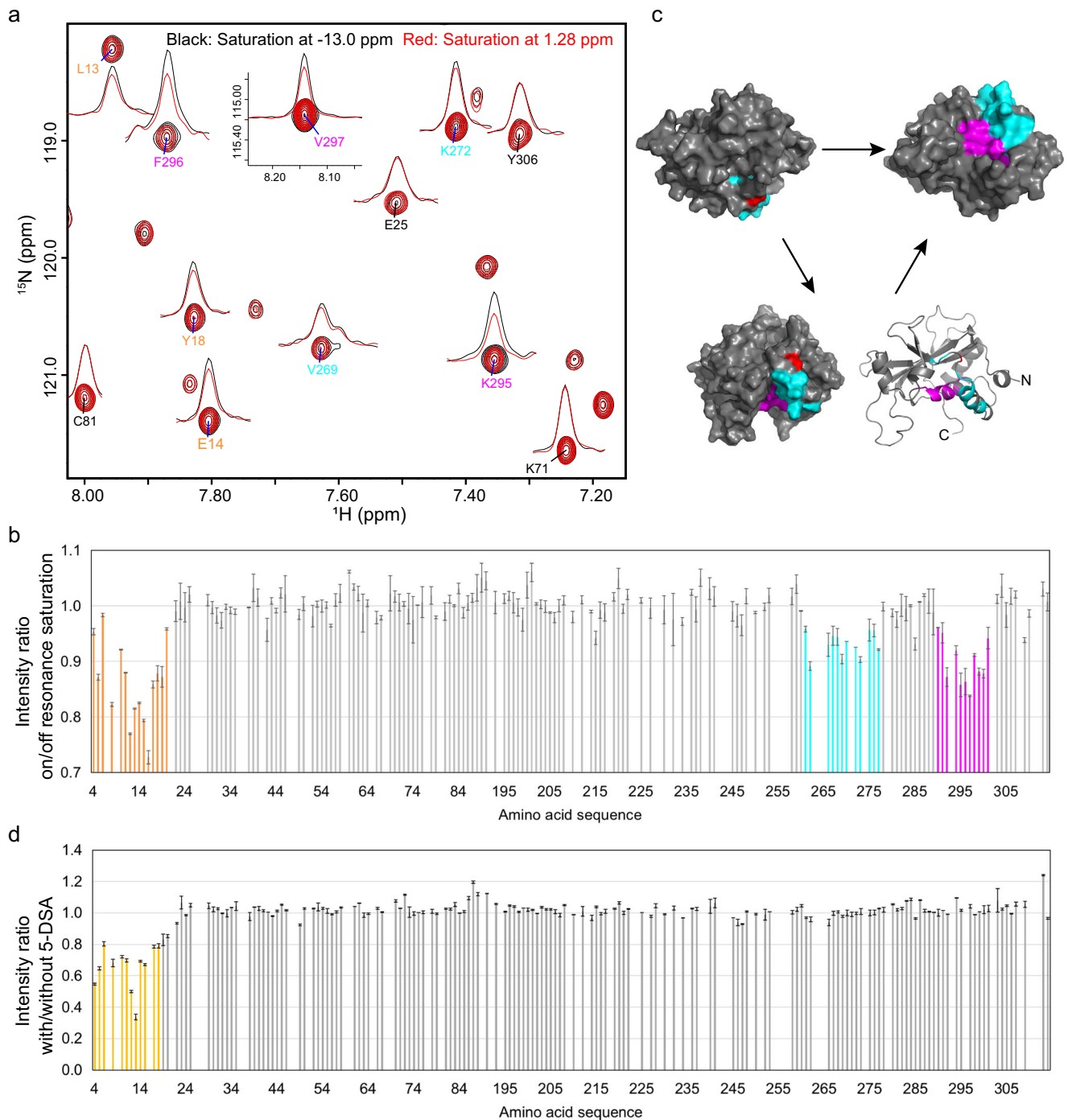

**Fig. 2 | Multifaceted membrane association of hAtg3$^{\Delta 90-190, 4M}$ determined by cross saturation and paramagnetic relaxation enhancement experiments.**
**a** Close-up of $^1$H-$^{15}$N TROSY spectra of 0.5 mM perdeuterated $^{15}$N, $^2$H- hAtg3$^{\Delta 90-190, 4M}$ in an 80% $D_2O$ and 20% $H_2O$ bicelles solution (DMPC:DMPG:DHPC = 8:2:20, q = 0.5, 12% w/v; 25 mM HEPES, pH 7.5, 75 mM NaCl, 2 mM TCEP.) with saturations at −13.0 ppm (black) and 1.28 ppm (red). **b** Plot of cross saturation effects against residue numbers. The perturbed residues from the NAH are colored orange, while those from 262 to 277 and 291 to 301 are colored cyan and magenta, respectively. Three

sets of $^1$H-$^{15}$N TROSY experiments with saturation on/off were collected and analyzed. **c** Perturbed residues from 262 to 277 (cyan) and 291 to 301 (magenta) are mapped onto hAtg3$^{\Delta N25, \Delta 90-190}$ NMR structure. Catalytic site Cys264 is indicated in red. **d** Plot of PRE effects from 5-DSA against residue numbers. A final concentration of 1.5 mM 5-DSA was included in the NMR sample of $^{15}$N, $^2$H-hAtg3$^{\Delta 90-190, 4M}$ in 12% (w/v) bicelles (DMPC:DMPG:DHPC = 8:2:20, q = 0.5). Two sets of $^1$H-$^{15}$N TROSY experiments with and without 5-DSA were collected and analyzed. For **b** and **d**, source data are provided as a Source Data file.

around the catalytic residue Cys264 (Fig. 2c). As a control, we performed the same experiment on a sample of perdeuterated $^{15}$N, $^2$H-hAtg3$^{\Delta 90-190, 4M}$ in an 80% $D_2O$ and 20% $H_2O$ solution; all residues exhibited few cross-saturation effects as shown in Supplementary Fig. 5b. Additionally, as shown in Supplementary Fig. 6a, TROSY spectra of hAtg3 in bicelles with and without LPE show minor

perturbations that are ascribed to the general perturbations of bicelles by aliphatic chain of lysolipids since the spectra of hAtg3 in bicelles with LPE or LPC are nearly identical (Supplementary Fig. 6b). Therefore, these cross-saturation experiments provide tangible evidence that, in addition to the NAH, the C-terminal regions I and II of hAtg3 also directly interact with the membrane.

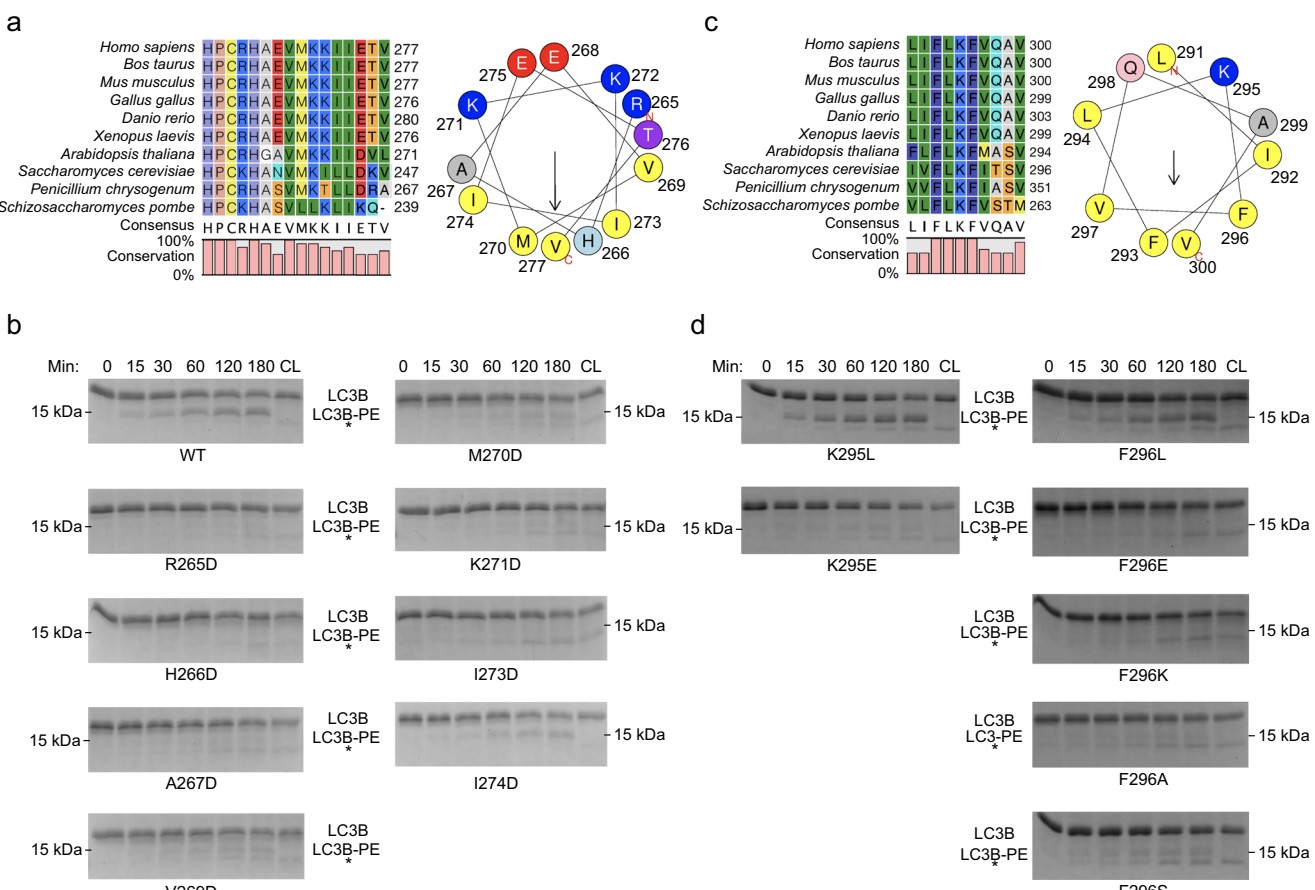

**Fig. 3 | hAtg3 C-terminal interaction with the membrane is indispensable for LC3B conjugation in vitro. a** Left panel: sequence comparison of hAtg3 C-terminal residues 262–277 with the analogous region in selected organisms. Right panel: helical wheel plot for hAtg3 residues 265–277. **b** Representative SDS-PAGE gel images of time-dependent formation of LC3B-PE for hAtg3 wildtype (WT) and mutants (n = 3). CL is the control incubated for 180 min without liposomes. Asterisk indicates a small amount of degradation of LC3B in the presence of ATP. **c** Left panel: sequence comparison of hAtg3 C-terminal residues 291–300 with the analogous region in selected organisms. Right panel: helical wheel plot for hAtg3 residues 291–300. **d** Representative SDS-PAGE gel images of time-dependent formation of LC3B-PE for hAtg3 mutants (n = 3). CL is the control incubated for 180 min without liposomes. Asterisk indicates a small amount of degradation of LC3B in the presence of ATP. For **b** and **d**, source data are provided as a Source Data file.

To probe the insertion depth of these membrane-interacting regions, we performed NMR paramagnetic relaxation enhancement (PRE) experiments using 5-doxylstearic acid (5-DSA)[48–51], a membrane-soluble probe. As shown in Fig. 2d, residues in the hAtg3 NAH exhibit large PREs, while all other residues experience few or no effects. Since the PRE probe is embedded in the acyl chains of the lipid bilayer, these results indicate that the NAH inserts into the hydrophobic core of the membrane. Together, our NMR chemical shift perturbation, cross-saturation, and PRE experiments demonstrate that hAtg3 interacts with the membrane using multiple surfaces. In addition to the NAH that inserts deeply into the membrane and interacts with the hydrophobic core, we postulate that C-terminal regions I and II interact with the polar headgroup layer of the membrane. Importantly, these hAtg3 C-terminus/membrane interactions depend on the membrane binding of its NAH, but not vice versa. As shown in our previous study, the hAtg3$^{\Delta 90–190,V8D\_V15K}$ mutant does not bind to the membrane, and few CSPs are observed for C-terminal residues (Supplementary Figs. 5 and 15(b)[30]). By comparison, hAtg3 mutants that disrupt this C-terminal membrane interaction still bind to the membrane, as described below.

### Mutational analyses of membrane-interacting regions I and II of hAtg3

We next examined the nature of protein-lipid interactions of the two C-terminal regions identified in the cross-saturation experiment.

Residues 262–277 of region I are highly conserved (Fig. 3a), and secondary chemical shift analyses of their assigned $^{13}C_{\alpha}$ and $^{13}C_{\beta}$ shifts in bicelle-bound hAtg3$^{\Delta 90–190,\ 4M}$ indicate that residues 265 to 277 are in a helical conformation (Supplementary Fig. 7). Interestingly, when plotted on a helix wheel these residues segregate into a hydrophobic or a hydrophilic face (Fig. 3a). The polar face includes residues Arg265, Glu268, Lys271, Lys272, Glu275, and Thr276 and the non-polar face includes residues His266, Ala267, Val269, Met270, Ile273, Ile274, and Val277. Thus, residues 265–277 form an amphipathic helix upon membrane binding. Notably, some of these hydrophobic amino acids are exposed to solvent in the apo structure of hAtg3 and are presumably primed to interact with the membrane. Consistent with this hypothesis, as shown in Fig. 3b, the substitution of these hydrophobic residues with a negatively charged Asp abolishes (H266D, A267D, and I273D) or markedly reduces (V269D and M270D) LC3B-PE formation in an in vitro conjugation assay. The I274D mutant showed a modest reduction in LC3B-PE formation, likely because Ile274 locates at the border of hydrophobic and hydrophilic faces. In addition, the substitution of positively charged Arg or Lys residues that presumably locate at the membrane/solution interface to a negatively charged Asp (R265D and K271D) also markedly reduces or disrupts the formation of LC3B-PE. As a control, we confirmed that these mutants form the LC3B-hAtg3 intermediate normally (Supplementary Fig. 8).

For residues 291–300 of region II, secondary chemical shift analyses of their assigned $^{13}C_\alpha$ and $^{13}C_\beta$ shifts in bicelle-bound hAtg3$^{\Delta90–190, 4M}$ indicate that these residues adopt a helical conformation (Supplementary Fig. 7), similar to the apo structure. This helix is comprised of predominantly hydrophobic residues; a helix wheel plot shows a very large non-polar face and a small polar face containing only Lys295 and Gln298 residues (Fig. 3c). In region II, there are four consecutive residues (F$^{293}$LKF$^{296}$) that are fully conserved and experience large cross-saturation effects (Fig. 2a, b). Phe293 and Leu294 interacts with strands of the central β-sheets, and their mutations would likely disrupt hAtg3's structure. In fact, the F293S mutation was not stable. To examine the interactions of Lys295 and Phe296 with the membrane, we constructed single site substitutions. To test whether Lys295 is embedded into the membrane while extending or "snorkeling" its positively charged sidechain to interact with the negatively charged phospholipid head groups, we substituted Lys295 with a Leu or a Glu residue. The hAtg3$^{K295L}$ mutant remained fully functional, but hAtg3$^{K295E}$ nearly completely lost its conjugase activity (Fig. 3d). Similarly, while the hAtg3$^{F296L}$ mutant retained its enzymatic activity, replacing Phe296 with an acidic Glu (F296E), a polar Ser (F296S), or a small hydrophobic Ala (F296A) residue nearly abolished its activity. Again, these mutations form the LC3B-hAtg3 intermediate and bind to the membrane normally (Supplementary Figs. 8 and 9). Thus, our mutagenesis results clearly support the association of two C-terminal regions of hAtg3 with the membrane.

The functional importance of Phe296 was recognized in a previous study of yAtg3[52]. Substitution of the corresponding Phe residue with a Ser (F293S) in yAtg3 was reported to dramatically increase its conjugase activity. This result is inconsistent with our finding that hAtg3$^{F296S}$ is impaired in LC3B-PE conjugation, suggesting that yAtg3 and hAtg3 may have subtly different catalytic mechanisms. To further investigate this, we introduced wild-type hAtg3 and the hAtg3$^{F296S}$ or hAtg3$^{F296L}$ variant into Atg3$^{-/-}$ mouse embryonic fibroblasts (MEFs) by lentiviral transduction (Fig. 4a) and examined the LC3B-PE conjugation in vivo by western blot analysis. While the LC3B-PE conjugate (LC3B-II) was absent in control Atg3$^{-/-}$ MEFs transduced with an empty vector (EV), the expression of hAtg3 rescued the LC3B-PE conjugation and autophagic flux (Fig. 4b–d). Consistent with our in vitro results (Fig. 3d), the introduction of the hAtg3$^{F296S}$ variant into Atg3$^{-/-}$ MEFs failed to rescue LC3B-II production and autophagic flux. In contrast, the hAtg3$^{F296L}$ mutant retained its ability to induce LC3B lipidation and autophagic flux to a similar extent as hAtg3 (Fig. 4b–d).

## Discussion

During autophagy the Atg3 enzyme catalyzes a covalent conjugation of Atg8 family proteins to the amino group of PE lipids. The reaction product Atg8-PE is required for the successful formation of an autophagosome. Since this reaction is carried out at the membrane surface with PE serving as one of two reaction substrates, the functional importance of the interaction between Atg3 and membrane has been long recognized. In an early study of yAtg3, PE lipids were reportedly required to anchor in a bilayer structure for Atg8-PE conjugation to occur, and the conjugation reaction required the presence of negatively charged lipids and proceeded more efficiently with a higher percentage of PEs in the membrane (up to 75% percent)[24]. Similarly, bilayer-like bicelles supported hAtg3 conjugase activity whereas DPC micelles did not[30]. Furthermore, the Atg3/membrane interaction mediated by its curvature-sensitive NAH has been well-established[27–29]. We recently demonstrated that the membrane binding of hAtg3 NAH is necessary but not sufficient for LC3-PE conjugation; we also uncovered a conserved region in the N-terminal of hAtg3 that works together with the NAH to couple its curvature-selective membrane binding to conjugase activity[30]. Building on this, here, using NMR CSPs, cross-saturation, and PRE experiments we reveal that, in addition to its NAH, the catalytically important C-terminal regions of

hAtg3, including the loop that harbors the catalytic residue Cys264 and two C-terminal helices (F and G), directly interact with the membrane. The functional relevance of these interactions is demonstrated by in vitro conjugation and in vivo cellular assays. Intriguingly, the catalytic loop and helix F of hAtg3 are conformationally flexible in aqueous solution as evidenced by resonance exchange broadenings and fast H/D exchanges. This observation further supports the model that they are conformationally plastic and susceptible to structural rearrangements that are critical for their conjugase activity[53].

We recently reported substantial chemical shift perturbations in the C-terminal regions surrounding the catalytic residue Cys264 of hAtg3 when its NAH binds to the membrane[30]. However, in our previous study, we could only obtain ~78% of backbone resonance assignments of bicelle-bound hAtg3$^{\Delta90–190}$ because of exchange broadenings[42]. Here, we overcame this challenge by introducing mutations (e.g. H266L mutation) that stabilize hAtg3's interaction with the membrane and achieved near-complete (~93%) backbone resonance assignments for bicelle-bound hAtg3$^{\Delta90–190, 4M}$. Consistent with our previous study, perturbed residues, induced by NAH binding to the membrane, map to a continuous surface around residue Cys264 and are located on one side of the protein. Moreover, NMR cross-saturation experiments provided evidence that, besides the NAH, the catalytic loop and two following helices (F and G) also directly interact with the membrane. These C-terminal/membrane interactions are functionally indispensable; mutations that are designed to disrupt these interactions abolish or dramatically reduce LC3-PE conjugation in vitro and in vivo. We propose a model that summarizes the key findings of this study and delineates the process of hAtg3/membrane interaction (Fig. 5). In the apo state, hAtg3 assumes an inactive conformation with an unstructured NAH and conformationally flexible catalytic loop and helix F. This structural plasticity of hAtg3 may be functionally advantageous for substrate loading and unloading. At highly curved membrane surfaces, the NAH rearranges into an amphipathic helix and inserts into the membrane; the catalytic loop and two adjacent C-terminal helices, F and G, subsequently locate to and interact with the membrane surface. We postulate that this multifaceted membrane association of hAtg3 mechanistically brings the substrates into proximity of the catalytic site, optimally orienting them via structural arrangements necessary for efficient LC3-PE conjugation. Recently, a coincident association of PX and C2 domains with the membrane was shown to be a major trigger for PI3KC2α activation[54] and a multivalent association of the PHLEHA7 PH domain with phosphatidyl-inositol-phosphate (PIP) induced PIP clustering[55]. In addition, during the activation of phospholipase A2 (PLA2), the membrane may function as an allosteric ligand[56].

Future studies, in particular structure determinations of the LC3-hAtg3 intermediate in the absence and presence of the Atg12-Atg5/Atg16 complex at the membrane surface, will illustrate how the multifaceted membrane association of NAH and C-terminus of hAtg3 tightly coordinates its curvature sensitivity and conjugase activity. In a recent study of yAtg3[43], a structural model of the Atg8-yAtg3 intermediate was proposed using molecular modeling in conjunction with mutagenesis and NMR CSPs. In this model, Atg8-yAtg3 adopts a closed-state, that is presumably more active for discharging Atg8 to PE lipids than an open-state, and a noncovalent interaction interface between yAtg3 and Atg8 is defined. Assuming that this interaction interface is conserved between yAtg3 and hAtg3, we have mapped the corresponding residues at the Atg8-yAtg3 interface onto the hAtg3 structure. As shown in Supplementary Fig. 10a, most of these residues locate outside of the major membrane binding regions (the catalytic loop, helices F and G). Therefore, these regions in LC3B-hAtg3 remain available to interact with the bilayer. Furthermore, in a preliminary study of LC3B-hAtg3 conjugated through a disulfide bond in the absence and presence of bicelles, residues showing substantial CSPs include those from the N-terminus (e.g. Val15, Leu19, Val22, and Leu23/

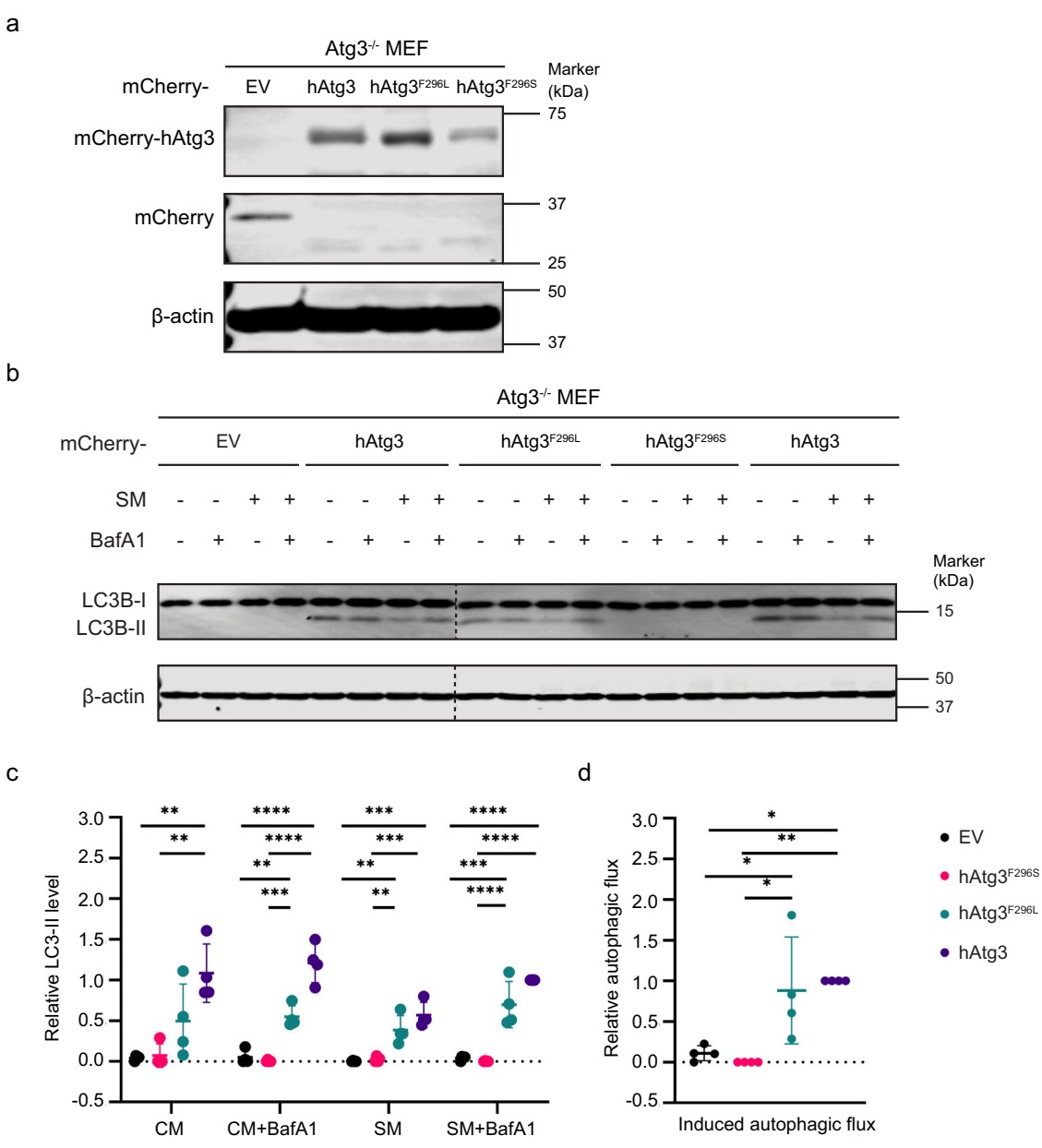

**Fig. 4 | Interaction of hAtg3 F296 residue with the membrane is indispensable for LC3B conjugation and autophagic flux in vivo. a** Atg3 knockout (Atg3$^{-/-}$) mouse embryonic fibroblasts (MEFs) stably expressing mCherry-EV (empty vector), mCherry-hAtg3, mCherry-hAtg3$^{F296L}$, or mCherry-hAtg3$^{F296S}$ were cultured in complete media with 100 nM bafilomycin A1 for 3 h and subjected to immunoblotting for mCherry. **b** Representative immunoblot (*n* = 4 blots). MEF cells were cultured in complete media (CM) or starvation media (SM) with/without 100 nM bafilomycin A1 (BafA1) to block LC3B-II (LC3B-PE) degradation for 3 h and subjected to immunoblotting with indicated antibodies. **c** Quantitative analysis of the relative LC3B-II level (*n* = 4 blots) in in vivo LC3B lipidation experiments. **d** Quantitative analysis of induced autophagic flux (*n* = 4 blots). Statistical analysis was performed using one-way ANOVA test followed by Turkey's multiple comparisons test. In **c**, data are presented as mean ± SD, *P* values: ****$P$ < 0.0001, ***$P$ = 0.0001, 0.0002, 0.0001, 0.0001 (left to right, bottom to top); **$P$ = 0.0021, 0.0017, 0.002, 0.0052, and 0.0034 (left to right, bottom to top). In **d**, data are presented as mean ± SD, *P* values: *$P$ = 0.0125, 0.0283, and 0.0118 (bottom to top), **$P$ = 0.0053. Source data are provided as a Source Data file.

L13), helix F (e.g. Leu266, Val269, Ile273, and Val277), and helix G (e.g. Leu290, Leu291, Ile292, Leu294, Val297, Val300, and Ile301) (Supplementary Fig. 10b). In these experiments, because of the increased size of LC3B-hAtg3/bicelle complex chemical shifts of $^{13}$C-labeled CH$_3$ groups of Ile, Leu, and Val residues in a perdeuterated protein background were explored to characterize protein/lipid interactions. Compared to backbone $^{15}$N and H$^N$ groups, $^{13}$C chemical shifts of sidechain CH$_3$ groups are more specific to direct interactions. These results support our assumption that the multifaceted membrane interaction of hAtg3 uncovered in this study applies to the LC3B-hAtg3

intermediate, although details of the LC3B-hAtg3/lipid interactions may be additionally adjusted to accommodate the presence of LC3B. In contrast, little direct structural information is available for the LC3-Atg3 intermediate in complex with the Atg12-Atg5/Atg16. In yAtg3, a segment of fourteen residues (Ile129 to Lys142), referred to as Atg3$^{E123IR}$ (E1, E2, and E3-interacting region), was shown to interact with residues of helices F and G and functionally inhibit its conjugase activity in the apo state. Removing this auto-inhibition by the Atg3$^{E123IR}$/Atg12-Atg5 interaction was suggested to be a requirement for activating yAtg3. However, the corresponding residues of Atg3$^{E123IR}$ and its key

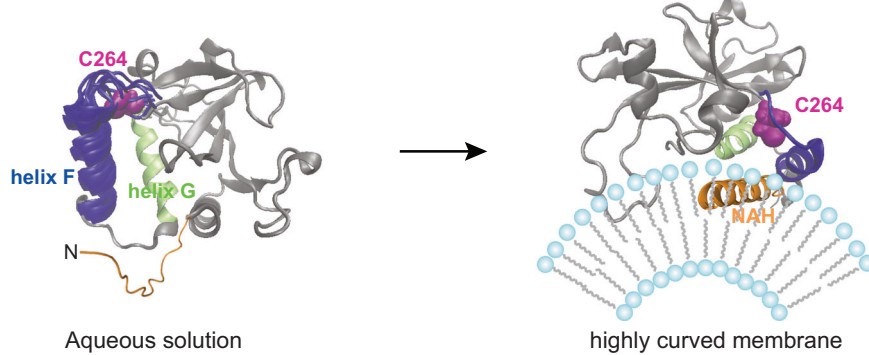

Aqueous solution                                          highly curved membrane

**Fig. 5 | A model delineates the process of hAtg3/membrane interaction.** In aqueous solution, hAtg3 has an unstructured NAH and conformationally flexible catalytic loop and helix F. At highly curved membrane surfaces, the NAH of hAtg3 rearranges into an amphipathic helix and inserts itself into the membrane; its catalytic loop and two C-terminal helices (F and G) subsequently locate to and interact with the membrane surface.

interacting partners on helices F and G are not conserved in hAtg3. Additionally, yAtg3 has an active AIM/LIR motif, while higher organisms have evolved to lose the LIR from Atg3. Therefore, the Atg12-Atg5/Atg16 complex may use different molecular mechanisms to activate yAtg3 and hAtg3 for Atg8-PE and LC3-PE conjugations. Nonetheless, the plasticity of Atg3's catalytically important C-terminal regions and their versatile interactions with the membrane, LC3, and the Atg12-Atg5/Atg6 complex (direct or indirect) are keys to the activation and regulation of its conjugase activity.

In summary, in this study we determined the solution structure of hAtg3 and demonstrated that its catalytic loop and adjacent helix F are conformationally dynamic in aqueous solution. Moreover, we identified direct interactions between its catalytically important C-terminal regions and the membrane. The functional implications of these hAtg3 C-terminal/membrane interactions were confirmed by mutagenesis and in vitro and in vivo assays. These interactions functionally link hAtg3 conjugase activity to NAH curvature sensitivity, thereby restricting LC3-PE conjugation to the highly curved rim of the phagophore. Thus conceptually, the membrane functions like an-E3 ligase for LC3-PE conjugation.

## Methods

### Mammalian cell culture
HEK293T cells (ATCC; CRL-3216) and Atg3$^{-/-}$ mouse embryonic fibroblasts (MEFs) provided by Dr. Shengkan (Victor) Jin (Rutgers University -Robert Wood Johnson Medical School, NJ) were cultured in Dulbecco's Modification of Eagle's Medium (DMEM) supplemented with 10% fetal bovine serum and 1× Antibiotic Antimycotic Solution (AA) (Cytiva, SV30079.01).

### Plasmid construction for in vivo functional assay
hAtg3, hAtg3$^{F296S}$, and hAtg3$^{F296L}$ cDNAs were amplified by PCR (forward: 5'-GAGCTGTACAAGTCTAGAGTGATGCAGAATGTGATTAATA-3'; reverse 5'-CGCAGATCCTTGCGGCCGCGTTACATTGTGAAGTGTCTTG-3'). Purified PCR products were sub-cloned into pCDH1-CMV-mCherry-MCS-EF1-puro viral backbone using NheI and BamHI.

### Lentiviral packaging, transduction, and cell sorting
Lentivirus was made by transfecting HEK293T cells with ViraPower packaging plasmids (pLP1, pLP2, and pLP/VSVG) from Invitrogen (K497500) and the viral vector using JetPrime (Polyplus, 101000015). Four hours after transfection, the culture medium was replaced. The viral supernatant was collected at 24 and 48 h post-transfection, and filtered with 0.45 μM supor membrane (Pall, 4614). The viral supernatant was added to Atg3$^{-/-}$ MEFs and incubated for 24 h. Puromycin selection was started after 48 h post transduction and remained for 3 days. MEF cells were sorted with flow cytometry (BD biosciences, BD

FACSAria Special Order Research Product (SORP), v2) based on mCherry expression level. MEF cells with high expression of mCherry were used for in vivo experiments.

### In vivo LC3-II synthesis measurement
Cells were incubated in complete or starvation media in the absence and presence of 100 nM bafilomycin A1 (BafA1) (ThermoFisher scientific, J61835.MCR) for 3 h and subjected to immunoblotting using the mCherry antibody (Abcam, ab125096) with a ratio of 1:1000, the β-actin antibody (Sigma, A5441-100UL) with a ratio of 1:10,000, and the LC3 antibody (Novus Biologicals, NB100-2220) with a ratio of 1:5000. To quantify the LC3-II expression level, western blot experiments were performed 4 times. The signal intensity of LC3-II was measured using Image Studio version 5 software (LI-COR Biotechnology), then normalized to the intensity of β-actin of the same sample. To calculate the relative LC3-II level, the LC3-II value of Atg3$^{-/-}$ MEFs expressing mCherry-hAtg3 in starvation medium with BafA1 was set to 1. The autophagic flux was calculated by subtracting the LC3-II value in the absence of BafA1 from that in the presence of BafA1 in starvation medium, and the values were then normalized with the value of Atg3$^{-/-}$ MEFs expressing mCherry-hAtg3 being set as 1. The data were analyzed by one-way ANOVA test followed by Turkey's multiple comparisons test using GraphPad Prism7.0.

### Protein expression and purification
Full-length human Atg3 (hAtg3), subcloned into the pET28a expression vector at the BamHI/XhoI site with His- and T7-tags at the N-termini, was kindly provided by Dr. Xuejun Jiang (Memorial Sloan Kettering Cancer Center, New York). For the studies here, a thrombin cleavage site was introduced at just before the start of hAtg3 N-terminus (with two amino acid residues Gly-Ser left with the rest of hAtg3 after thrombin cleavage), or hAtg3 without the N-terminal first 25 residues (with only one Gly left with the rest of hAtg3 without the N-terminal first 25 residues after thrombin cleavage). These constructs as well as hAtg3 without residues 90–190 were obtained previously using primers listed in Supplementary Table 2[30]. All other hAtg3 mutants were generated using the Q5 Site-Directed Mutagenesis Kit (New England Biolabs, E0554S) or QuikChange Multi Site-Directed Mutagenesis Kit (Agilent Technologies, 200515-5) with corresponding primers (listed in Supplementary Table 2) and verified by sequencing. Plasmids containing target genes were transformed into chemically competent Rosetta™(DE3) pLysS cells (Novagen, 70956) for expression. Typically, a single colony was selected to grow in a small volume of LB medium overnight at 37 °C as a starter culture and then inoculated into a large volume of LB medium (for unlabeled proteins) or M9 medium supplemented with D-glucose (3 g/L, or D-glucose-$^{13}$C$_6$ (Cambridge isotope laboratories, CLM-1396) and D$_2$O (Cambridge isotope laboratories, DLM-4-99.8; or Sigma-Aldrich, 756822)) and $^{15}$NH$_4$Cl

(1 g/L, Cambridge isotope laboratories, NLM-467; or Sigma-Aldrich, 299251) for $^{15}N$ (or $^{15}N/^{13}C/^2H$, or $^{15}N/^2H$) labeled samples at 37 °C until $OD_{600}$ reached 0.6. The temperature was then lowered to 25 °C and cells were induced with 0.5 mM IPTG for ~16 h and then harvested by centrifugation. For $^{13}CH_3$-ILV-labeling, a single colony was cultured in 5 mL LB medium for 7 h. Cells from 0.2 mL of the culture were pelleted down, washed with 0.5 mL M9 $D_2O$ medium once, and then inoculated into 10 mL fresh M9 $D_2O$ medium at 37 °C for 15 h. The culture was then transferred into 400 mL fresh M9 $D_2O$ medium and incubated at 37 °C until $OD_{600}$ reached 0.6. The temperature was then lowered to 25 °C, α-Ketobutyric acid-4-$^{13}$C,3,3-$D_2$ (Cambridge isotope laboratories, CDLM-7318, 70 mg/L) and 2-Keto-3-methyl-$^{13}$C-butyric-4-$^{13}$C,3-D acid (Sigma-Aldrich, 691887, 120 mg/L) were added to the growth medium. After incubation for 1 h, cells were induced with 0.5 mM IPTG for ~16 h.

Cell pellets containing expressed proteins were suspended in a lysis buffer of 20 mM phosphate, pH 7.5, 300 mM NaCl, 2 mM beta-mercaptoethanol (BME), and 1 mM $MgCl_2$ supplemented with benzonase nuclease (Millipore, 70746) and complete protease inhibitor cocktail (Roche, 11836170001). The cells were lysed by sonication on ice with 2 s on and 7 s off intervals for 18 min total duration. Cell debris was removed by centrifugation (Sorvall RC5B Plus Refrigerated Centrifuge) at 26,900 × $g$ at 10 °C for 30 min. Supernatants were collected, and loaded onto a Ni-NTA column (HisTrap HP, Cytiva 17524801). The column was washed with PBS buffers (20 mM phosphate, pH 7.5, 300 mM NaCl, 2 mM BME) without and with 30 mM imidazole, and then eluted with PBS buffer containing 500 mM imidazole. The elute of hAtg3 and its mutants (with purities no less than 90% after analysis on an SDS-PAGE gel) were concentrated for biochemical assays. For NMR experiments, the elute of proteins from the Ni-NTA column was concentrated and exchanged to a buffer containing 50 mM HEPES, pH 7.5, 150 mM NaCl, and 2 mM BME, followed by the addition of 0.1% (v/v) TWEEN20 (Fisher Scientific, BP337-500) and 50 units thrombin (Sigma-Aldrich, 605157) to remove the T7- and His-tags for an overnight agitation at 4 °C. The solution was then subjected to a Q-column (HiTrap Capto$^{TM}$ Q, Cytiva 11001302) and further purified by size-exclusion chromatography using an S200 column (HiLoad 16/60 Superdex 200, Cytiva 28-9893-35) and a buffer of 50 mM HEPES, 1 M NaCl, and 1 mM DTT. LC3B and mouse Atg7 for in vitro assays were prepared as described previously[30]. Purified proteins were exchanged into a buffer of 50 mM HEPES, pH 7.5 (for functional assays, NMR in bicelles and aqueous solution) or pH 6.5 (for NMR structure determination in aqueous solution), 150 mM NaCl, and 2 mM TCEP (tris(2-carboxyethyl)phosphine). Protein concentration was assayed using a Nanodrop (ND-1000 Spectrophotometer, Thermo Scientific, 2353-30-0010).

## In vitro conjugation assay

In vitro LC3B-PE conjugation was conducted as previously described[30]. Typically, 5.0 µM hAtg3 or its mutant was mixed with 5 µM LC3B, 0.5 µM mouse Atg7, 1 mM $MgCl_2$ in a reaction buffer (50 mM HEPES, 150 mM NaCl, 2 mM TCEP, and pH 7.5) of 37 µL. 3.7 µL were removed for a control. Then 1 mM ATP (0.45 µL of 100 mM ATP) was added, and the reaction was allowed to proceed for 30 minutes at 37 °C for intermediate formation. 3.75 µL were removed to check for intermediate formations, and an additional 3.75 µL were incubated separately as a non-liposome containing control (CL). 1 mM sonicated liposomes (8.75 µL of 4 mM stock, POPC/DOPG/DOPE with a molar ratio of 3:2:5) were then added to the reaction system (final volume of 35 µL). The reaction proceeded at 37 °C, with 5 µL aliquots removed at 0, 15, 30, 60, 120, and 180 minutes. Samples were mixed with 4 µL 4× protein loading buffer (10% w/v SDS, 10% BME, 40% Glycerol, 250 mM Tris−HCL, pH 6.8, and 0.4% w/v bromophenol blue dye), and stored at −20 °C until analyzed by 18% polyacrylamide gel electrophoresis after heating for 3 min. 4× loading buffer without BME was used for the gel analysis of intermediates. Gels were imaged on a BioRad

Chemidock MP imager, and analyzed using ImageJ. For the preparation of sonicated liposomes, POPC (Avanti Polar Lipids, 850457), DOPG (Avanti Polar Lipids, 840475), DOPE (Avanti Polar Lipids, 850725) in chloroform were added to a glass tube and dried to a thin film by spinning with heat for half an hour using a condenser rotor SpeedVac, followed by lyophilization overnight once the volatile organics were removed. Lipids were rehydrated with $H_2O$ for 1 h at 42 °C, with vortex and then freeze at −80 °C every 15 min. The mixture was then for bath sonication (BRANSON 3510R-MT Bransonic Ultrasonic Cleaner). The lipids were sonicated for 4 × 15-min intervals until clear, then stored at room temperature.

## Liposome co-sedimentation assay

Liposome co-sedimentation assay was conducted as previously described[30]. Typically, 2 µM hAtg3 (or its mutants) was incubated with 800 µM sonicated liposomes (POPC/DOPG/DOPE with a molar ratio of 3:2:5) or without liposomes as a control in a buffer A of 50 mM HEPES, 150 mM NaCl, and 2 mM TCEP at 37 °C for 1 h. After incubation, 12 µL from a total of 300 µL mixture were removed and mixed with 4 µL 4× protein loading buffer (total, T). The remaining mixture was centrifuged at 160,000 × $g$ for 2 h at 4 °C (Optima MAX Ultracentrifuge). After removing the supernatant, the pellets were gently washed once with 288 µL buffer A and resuspended in 288 µL buffer A. Then, 12 µL of the supernatant (S) and the pellet resuspension (P) were mixed with 4 µL 4× protein loading buffer, respectively, for gel electrophoresis (NuPAGE 10% Bis-Tris, Invitrogen, NP0303BOX; or SurePAGE 10% Bis-Tris, GenScript, M00666). Samples were heated at 95 °C for 3 min for gel electrophoresis.

## Alignment media preparation

For the RDC experiment, we prepared charged polyacrylamide gels as described previously[57]. Briefly, a stock solution of 40% neutral acrylamide and N, N′-methylenebisacrylamide in a 19:1 ratio (Bio-Rad, #161-0144) was mixed with a stock solution of 40% positively charged (3-acrylamidopropyl)-trimethylammonium chloride (75% wt, Sigma-Aldrich, 448281) or negatively charged 2-acrylamido-2-methyl-1-propanesulfonic acid (AMPS, Sigma-Aldrich, 282731) containing N,N′-methylenebisacrylamide (Sigma-Aldrich, M7279) in a 19:1 ratio in equal volume. The mixtures or the stock solution containing only neutral acrylamide were diluted with 10× TBE buffer (Invitrogen, 15581-044) to a final 5% acrylamide concentration for the negatively-charged or neutral gel, or a final 7% acrylamide concentration for the positively-charged gel. Polymerization was initiated by adding 0.1% ammonium peroxide sulfate and 1% tetramethylethylenediamine (TEMED, Bio-Rad, #161-0800). The polymerization of 130 µL of the mixtures was carried out in a 3.2 mm diameter plastic tube. Polymerized gels were extensively washed in deionized water, and the water was changed at least 3 times within 2 days. The gels were dried over a 2-day period at 37 °C on a plastic support wrapped with polyvinylidene chloride (PVDC) foil. The dried gel was transferred to a 5 mm Shigemi tube for the NMR sample, and the protein solution was added. Vertical compression of the gel was achieved by inserting the Shigemi plunger into the tube and limiting the final length at around 14 mm. The sample was ready for NMR measurements after equilibration for at least 8 h at 4 °C. For a phage aligned NMR sample, 500 µM $^{15}N$, $^2H$-labeled protein was dissolved in 50 mM HEPES, pH 6.5, 150 mM NaCl and 2 mM TCEP containing 6.5 mg/mL phage Pf1 (ASLA BIOTECH, P-50-P).

## hAtg3 Cys264 and LC3B G120C disulfide cross-linking

LC3B$^{G120C}$- hAtg3$^{Δ90−190, H266L, C50A, C81A, C85A, C259A}$ linked by a disulfide bond was prepared following a published protocol[43]. Briefly, 500 µM unlabeled LC3B$^{G120C}$ was first reduced by 10 mM DTT and desalted into buffer B (25 mM HEPES, pH 7.5, 75 mM NaCl). Then this LC3B$^{G120C}$ protein solution was mixed with 1:1 v/v buffer C (25 mM HEPES, pH 7.5,

75 mM NaCl, 2.5 mM 2,2'-dipyridyldisulfide (DPS, Alfa Aesar, A11118)), incubated at room temperature (RT) for 20 min to form DPS-LC3B$^{G120C}$, and desalted again into buffer B. Separately, 300 μM perdeuterated $^{13}CH_3$-ILV-labeled hAtg3$^{Δ90–190, H266L, C50A, C81A, C85A, C259A}$ was reduced by 10 mM DTT and desalted into buffer B. Finally, this hAtg3 protein solution was mixed with DPS-LC3B$^{G120C}$ solution at a 1:1.5 molar ratio and incubated at RT for 1 h to form LC3B$^{G120C}$- hAtg3$^{Δ90–190, H266L, C50A, C81A, C85A, C259A}$. Cross-linking product was directly desalted into a buffer of 50 mM HEPES, pH 7.5, 150 mM NaCl and verified by SDS-PAGE gel with a purity of >90%.

## NMR spectroscopy

Typical NMR samples contained 0.5 mM labeled proteins in a buffer of 50 mM HEPES, pH 6.5, 150 mM NaCl, 2 mM TCEP, and 0.02% NaN$_3$. For H/D exchange experiments, 0.5 mM $^{15}N$-labeled proteins in 50 mM HEPES, pH 6.5, 150 mM NaCl, and 2 mM TCEP were lyophilized, and the powder was then dissolved in D$_2$O (or 4% D$_2$O/96% H$_2$O, for reference) for immediate NMR collection. For bicelle samples, typically 0.5 mM $^{15}N$, $^{15}N/^{13}C$, or $^{15}N/^{13}CH_3$-ILV $^2$H-labeled proteins were mixed with a final concentration of 12% (w/v) bicelles (DMPC:DMPG:DHPC = 4:1:20 (molar ratio, $q = 0.25$), or DMPC:DMPG:DHPC = 8:2:20 (molar ratio, $q = 0.5$, for cross saturation and PRE 5-DSA experiments), or DMPC:DMPG:LPE:DHPC = 7:2:1:20 (molar ratio, $q = 0.5$), or DMPC:DMPG:LPC:DHPC = 7:2:1:20 (molar ratio, $q = 0.5$) in 25 mM HEPES, pH 7.5, 75 mM NaCl, 2 mM TCEP. Bicelles were prepared as previously described[30]. Briefly, DMPC (Avanti Polar Lipids, 850345), DMPG (Avanti Polar Lipids, 840445), or LPE (Avanti Polar Lipids, 856735), or LPC (Avanti Polar Lipids, 855575) were mixed with double distilled H$_2$O (ddH$_2$O), frozen at −80 °C, and thawed at 42 °C for 3 freeze-thaw cycles. DHPC (Avanti Polar Lipids, 850305) in ddH$_2$O was then added into the mixture and quickly vortexed to form a clear solution. The solution was frozen at −80 °C and slowly thawed at room temperature for use. For PRE experiments, 5-DSA (Sigma-Aldrich, 253634) powder was dissolved in 24% (w/v) bicelles as a stock solution (100 mM), and titrated into NMR samples to reported concentrations; the mixture was vortexed, frozen at −80 °C, and thawed at room temperature for NMR data collections.

All NMR data were acquired at 25 °C on a Bruker 600 MHz spectrometer except the NOESY data that were acquired on a Bruker 850 MHz spectrometer. Both instruments are equipped with cryoprobes. The data were processed using NMRPipe3.0 and analyzed using NMRViewJ (Version 9.1.0-b55 with Java 1.8.0). Backbone resonance assignments were carried out using triple resonance HNCO, HN(CA)CO, HNCA, HN(CO)CA, HNCACB, and HN(CO)CACB experiments (Bruker Topspin3.2 pulse sequence library). Aliphatic and aromatic side-chain resonance assignments were obtained from the 3D $^{13}C$-edited HCCHTOCSY, $^{13}C$-edited NOESY, HCACO, $^{15}N$-edited NOESY and HBHA(CO)NH, 2D Hb(CbCgCd)Hd and Hb(CbCgCdCe)He spectra (Bruker Topspin3.2 pulse sequence library). The 3D $^{13}C$-NOESY and $^{15}N$-NOESY experiments (Bruker Topspin3.2 pulse sequence library) were collected with 150 ms mixing time. $^1D_{NH}$ and $^2D_{C'H}$ RDCs for resonances with little perturbation between an aqueous solution and alignment medium were measured using the ARTSY method[58] and an IP-HSQC experiment[59], respectively.

Torsion angle restraints were derived from C$_α$, C$_β$, N, and C' chemical shifts using TALOS+. NOESY assignments were obtained by a combined manual and automated analysis with CYANA 3.0. The structures were calculated and refined with XPLOR-NIH 3.3. PROCHECK, PSVS, and PDB validation servers were used to analyze the ten lowest-energy structures with acceptable covalent geometry. In the final structures, the percentage of residues that reside in the most favored, additionally allowed, and generally allowed regions of the Ramachandran diagram are 90.6%, 9.3%, and 0.1%, respectively, for ordered residues (28–84, 196–261, 267–279, and 286–306).

## Statistics and reproducibility

Data are presented as mean values ± SD (standard deviation), calculated using GraphPad Prism 7.0. $P$ value < 0.05 was considered the threshold for statistical significance. The significance intervals (*) for $P$ values are provided within the figure legend, along with the specific statistical test performed for each experiment: one-way ANOVA test followed by Turkey's multiple comparisons test. The number of experimental replicates ($N$ values, $n$) is indicated within figure legends. The cell lines utilized were regularly authenticated through morphologic inspection and mycoplasma testing. To minimize clonal bias, the experiments were performed using pooled mCherry-positive cells sorted by flow cytometry.

## Reporting summary

Further information on research design is available in the Nature Portfolio Reporting Summary linked to this article.

## Data availability

NMR resonance assignments generated in this study have been deposited in the BMRB under accession codes 31065 and 51749. hAtg3 structure has been deposited in the Protein Data Bank under accession code 8FKM. Previously published NMR resonance assignment used in this study is available in the BMRB with accession code 50479. Previously published Atg3 structures used in this study are available in the PDB with accession codes: 6OJJ, 3VX8 and 2DYT. Source data are provided with this paper.

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

## Acknowledgements

F.T., H.G.W, and J.M.F. acknowledge the support from the National Institutes of Health through R01 GM127730 (F.T.), R01 GM127954 (H.G.W.), R01 CA222349 (H.G.W.) and Four Diamonds through 4D21_2024_1001 (F.T.) and 4DIA_153697 (J.M.F). The NMR instruments (SCR_023244) used in this project were funded, in part, by the Pennsylvania State University College of Medicine via the Office of the Vice Dean of Research and Graduate Students and the Pennsylvania Department of Health using Tobacco Settlement Funds (CURE). The content is solely the responsibility of the authors and does not necessarily represent the official views of the University or College of Medicine. The Pennsylvania Department of Health specifically disclaims responsibility for any analyses, interpretations, or conclusions.

## Author contributions

Y.S.Y., E.R.T., V.B., M.C.B., G.F.W., X.H., Y.S., H.G.W., and F.T. designed and performed the experiments and analyzed the data. Y.S.Y., V.B., M.C.B., H.G.W. and F.T. drafted figures and the manuscript. M.C.B. and J.M.F. helped with protein preparation, data analysis, and manuscript preparation. F.T. and H.G.W. conceived, planned, and supervised this study, and wrote the final paper with feedback from all authors.

## Competing interests

The authors declare no competing interests.
