## [Peer Review File · Nature Communications]

Multifaceted Membrane Interactions of Human Atg3 Promote LC3-Phosphatidylethanolamine Conjugation during AutophagyREVIEWER COMMENTS

Reviewer #1 (Remarks to the Author):

This work describes a significant advance in the mechanism of ATG3. The manuscript is a valuable continuation to the one published in 2021 in this journal (Ye et al 2021, doi: 10.1038/s41467-020-20607-0). The main innovation is that the catalytic region (or active center) is restructured when the protein becomes in contact with the membrane, through its N-terminal amphipathic helix (the role of this helix was already known). This result is reinforced by experiments in which residues deemed important are mutated, and the resulting AT3 mutants are tested in vivo in ATG3-knock out MEF cells.

The manuscript could be improved in a number of ways:

1. The NMR studies have been performed with an ATG3 construct in which residues 1-25 and 90-190 have been deleted, leaving the protein as hAtg3 Δ N25, Δ 90-190. Taking into account that native hAtg3 has 314 aa, removing 125 aa for the NMR studies may seriously damage the conclusions obtained. The authors should justify this very carefully. It is true that by removing those residues the protein remains practically identical to that of yeast, but the objective of the work is to study the human, not the yeast Atg3/ATG3.
2. Little attention is given to the lipidic aspects in the work. Bicelles are used, and from what is stated in the caption of Fig 2, these are composed of DMPC:DMPG:DHPC = 8:2:20. This description, lacking details on lipid:protein ratio, concentrations, or buffer in which bicelles are formed, is totally insufficient to allow reproduction of the experiments. Note that Nath et al (2014) and Hervas et al (2017) already saw that lipid composition and membrane curvature greatly influenced Atg3 activity, thus the authors must justify why they have chosen this composition and this system.
3. Again on the matter of lipid composition, there appears to be no PE in the bicelles used. Drawing conclusions from the behavior of ATG3 when binding to membranes in the absence of its main substrate PE, when it has been seen by several authors that, depending on the % PE, Atg3 activity can vary greatly, certainly limits the validity of the conclusions obtained by the authors. This point should be carefully discussed.
4. Checking the effect of the different mutations in vivo using Atg3-KO MEF provides very clear and consistent results with what was obtained in vitro. However, in addition to measuring basal autophagy by treating the cells with bafilomycin, it could have been tested how they behave in autophagy caused by other agents, e.g. starvation, treating the cells with nutrient-poor medium, or with rapamycin.

5. In the discussion section, one misses the role of the F and G helices when Atg3 interacts with the other components of the conjugation system, such as ATG7, ATG8 or the E3 complex. This would provide the reader with an idea on how the complete system works, and the importance of the F and G helices in Atg3 operation.

Reviewer #2 (Remarks to the Author):

The manuscript by Ye et al. (NCOMMS-23-03873, entitled “Translating Membrane Geometry into Protein Function: Multifaceted Membrane Interactions of Human Atg3 Promote LC3-Phosphatidylethanolamine Conjugation during Autophagy”) shed light on a very important and still not completely understood aspect of autophagosome biogenesis – molecular mechanism of Atg8 lipidation. Using NMR spectroscopy they were able to solve the solution structure of human ATG3 protein (its essential core compartments except the N-terminal amphiphilic α -helix) and demonstrate the conformational plasticity of the ATG3 C-terminal locus around catalytic Cys264. They also showed by rationally-designed mutagenesis that flexibility of the C-terminal region is dependent on the protonation state of the invariant His266, which appeared as a unique pH sensor in human ATG3. His266 protonation stabilizes inactive conformation of the ATG3, while deprotonation of His266 – in higher pH values or by H266L mutation – stabilizes active ATG3 conformation and switch on the catalytic E2 activity of the enzyme. Most interesting finding within the presented work is, in my opinion, that this flexible C-terminal part of the ATG3 is also interacting with membranes, organizing a close proximity of ATG3 catalytic Cys264, its substrate ATG8 and the second substrate, phosphatidylethanolamine (PE) near the highly curved phosphoric rim for effective ATG8-PE conjugation.

The manuscript is well written and clearly organized, cited literature is enough and sufficient to introduce readers into the subject. All experiments (NMR for structure determination and for evaluation of relative mobility of ATG3 regions, in-vitro reconstitution of Atg8 lipidation, and mutational analysis of the desired ATG3 residues) are at the high standards. The resulting conclusions are of sufficient novelty and of considerable biological interest, and in the discussion the authors convincingly point out that their work opens up several ways for follow-up studies geared at understanding the precise mechanisms of Atg8 lipidation. In this way, I found that the manuscript is suitable for publication in Nature Communication after some minor revisions.

Major point:

The obvious drawback of this study is that the obtained results and conclusions are based mostly on in-vitro experiments, limiting the understanding of real role of the discovered ATG3 C-terminal membrane-binding site on Atg8 lipidation. Connected to that, autophagy-related E3 complex

(ATG12~ATG5/ATG16L1) is essential for the effective in-vivo lipidation of ATG8-proteins. The authors performed in-vitro LC3 lipidation assays without E3, using ability of the Atg8 to conjugate lipids in presence of Atg7 (E1), Atg3 (E2), Atg8 and lipids (liposomes), reported from very first works on ATG3 enzymatic activity. The authors should at least discuss the relation between in-vitro and in-vivo catalytic activity of ATG3 and extrapolate conclusions on the complete Atg8-PE enzymatic cascade.

Minor points

1. Lines 49-50 – “lipidation ... to” is not correct statement, better to say “lipidation by” or “conjugation to”
2. Line 70 – “concurrently”? more precise description will be “sequentially”, as the later experiments shows that C-terminal ATG3 region interacts with the membrane only after the NAH is bound.
3. Line 74 – “substrates of LC3 and PE lipids” should be “substrates – LC3 and PE lipids”
4. Lines 113 and 127 – The deuterium exchange is not the method by which one can measure real stability, but rather flexibility of particular protein regions. I would recommend to avoid the term stability and state what helix F is more mobile rather than “less stable”
5. Line 138 – “colocalizes” is not the proper term here, better to state “is in close proximity”
6. Lines 163-164 and Extended Data Fig. 4b – seems, authors assumed “the CSP induced by bicelles”.
7. Line 202 – better “Analysis by mutagenesis” or “Mutational analysis”
8. Lines 207-208 – provide reference on the used method to predict secondary structure by Ca/Cb resonances values.
9. Line 337 – inability of ATG3 C-terminal region to interact with membrane without membrane association by N-terminal helix is one of the key result in the manuscript. I would additionally mention this in the Results section.
10. Lines 373-374 – difference between yeast Atg3 and human enzymatic mechanisms are supported by the fact that γ Atg3 has an active AIM/LIR motif, while higher organisms evolved to lose the LIR from Atg3.
11. Figures 3a, 3c and Extended Data Fig. 2 – will be better to have human protein sequences at the first row.
12. Figures 3b, 3d and 4a – indicate which time was used in CL (control probes w/o liposomes) in the figure legends.

Reviewer #3 (Remarks to the Author):

The authors of this study correlated structural characteristics of ATG3, revealed by NMR, with its function in autophagy. For NMR studies, a truncated variant of ATG3 lacking its N-terminus, which contains an amphipathic membrane binding helix, and a central part including aa 90-190, was used. This variant was designed based on the crystal structures of yeast Atg3 and Atg3 from *A. thaliana*, in both of which this region was not visible in the electron density. Not surprisingly, the solution structure of hATG3 superimposed very well with the crystal structures of yeast and plant Atg3. Importantly, the deleted region contained the ATG7 binding site, resulting in a strongly impaired catalytic activity of ATG3 delta90-190 to conjugate LC3 to PE. The authors reported a similar structure in their Nat. comm. paper in 2021. In this study, they expanded their characterization of ATG3 delta90-190 and determined exposed aa based on H/D exchange as well as membrane binding using bicelles. They found that the catalytic loop of ATG3 is highly dynamic and in close proximity with bicelles. In their previous publication, they already reported chemical shift perturbations near the catalytic site upon incubation of ATG3 delta90-190 with bicelles, but 44 residues remained unassigned. In this study, they optimized ATG3 delta90-190 by introducing four amino acid substitutions which stabilized ATG3 on bicelles and improved the NMR signals so that most aa could be assigned. Interestingly, this variant showed stronger chemical shift perturbations (CSPs) of residues in the C terminus of ATG3 compared to the previously reported CSPs of ATG3 delta90-190. The authors continued by performing cross-saturation experiments, revealing that besides residues of the N terminal amphipathic helix, residues that cluster around the catalytic site are in close proximity to lipids of bicelles. Secondary chemical shift analysis implied that residues 265-277 form a helix upon membrane binding with amphipathic character. However, as opposed to the N terminal amphipathic helix, this helix does not insert into the lipid bilayer. Mutating hydrophobic residues of that helix to aspartate interfered with LC3B lipidation in vitro. Residues of the second membrane binding site formed a largely hydrophobic helix and mutation of these residues either destabilized the protein or impacted on its catalytic activity. Notably, H266 was found to be pH sensitive and the interaction of ATG3 with membranes as well as its catalytic activity were abolished if H266 was protonated or mutated to Lysin, while mutation to leucine or phenylalanine rendered ATG3 pH insensitive, being active also under acidic conditions. However, the H266L or H266F mutants failed to complement ATG3 in ATG3KO MEFs, while H266K partially restored lipidation, which is in odds with observations made in vitro.

Overall, this study provides only incremental advances over to the previous study of the authors published in Nat. comm. in 2021. They expanded on the previous observation that residues near the catalytic site bind membranes, which would be expected because finally LC3 needs to be transferred to PE, which requires both entities to be in close proximity. Data that go beyond the previous study include identification of an amphipathic helix that does not insert into the membrane but contains a pH sensitive His that promotes LC3 lipidation only in its unprotonated state. What role this pH sensitivity plays during autophagy remains uncertain because the pH of the cytosol is with 7.4 in between the pH optimum of ATG3 (pH = 8.5) and the pH at which the activity of ATG3 is abolished (pH = 6). Moreover, there is a contradiction between the activity of ATG3 H266 mutants in vitro and in vivo, making it difficult to assign a specific function to H266 during LC3 lipidation.

Moreover, the study was conducted in the absence of LC3, which might have a profound impact on the interaction of the two helices F and G of ATG3 with membranes. Overall, the new data provided by this study does not merit publication in *Nat. Commun.* and the study appears to be too preliminary for publication in a more specialized journal. Notably, the contradiction of results from *in vitro* and *in vivo* experiments needs to be resolved and LC3 needs to be taken into account to reveal meaningful information on the orientation of the LC3-ATG3 conjugate on membranes. Finally, ATG12-5-16L1 also impacts on the recruitment and, presumably, the orientation of LC3-ATG3, arguing that the mechanism of ATG3 action can only be revealed if ATG3 is studied in the context of the E3-ligase and LC3.

Response to the Reviewers

We highly appreciate the reviewers for their encouraging and constructive comments. We considered them at length, endeavored to provide the requested experimental data and necessary clarifications, and revised the manuscript accordingly. As a result, we think the manuscript has become much stronger.

Reviewer #1 (Remarks to the Author):

This work describes a significant advance in the mechanism of ATG3. The manuscript is a valuable continuation to the one published in 2021 in this journal (Ye et al 2021, doi: 10.1038/s41467-020-20607-0). The main innovation is that the catalytic region (or active center) is restructured when the protein becomes in contact with the membrane, through its N-terminal amphipathic helix (the role of this helix was already known). This result is reinforced by experiments in which residues deemed important are mutated, and the resulting AT3 mutants are tested in vivo in ATG3-knock out MEF cells.

The manuscript could be improved in a number of ways:

1. The NMR studies have been performed with an ATG3 construct in which residues 1-25 and 90-190 have been deleted, leaving the protein as hAtg3 Δ N25, Δ 90-190. Taking into account that native hAtg3 has 314 aa, removing 125 aa for the NMR studies may seriously damage the conclusions obtained. The authors should justify this very carefully. It is true that by removing those residues the protein remains practically identical to that of yeast, but the objective of the work is to study the human, not the yeast Atg3/ATG3.

Author reply: The hAtg3 ^{Δ 90-190} construct was employed in this study to probe the hAtg3/bicelles interaction. We have previously shown that this construct remains functional *in vitro*, and deletion of residues 90 to 190 minimally perturbs the core structure of hAtg3 (Ye et al 2021, Nat. Commun., supplementary Figs. 6-8). To facilitate NMR structure determination in the absence of membranes, we used the hAtg3 ^{Δ N25, Δ 90-190} construct. In aqueous solution, residues 1 to 25 of hAtg3 are unstructured (Ye et al 202, Biomol. NMR Assign.). Their deletion causes a minimal perturbation in the core structure (supported by an overlay of the TROSY spectra of hAtg3 ^{Δ 90-190} and hAtg3 ^{Δ N25, Δ 90-190} shown in Extended Data Fig. 1). We have revised the text to justify using these constructs and added a reference.

2. Little attention is given to the lipidic aspects in the work. Bicelles are used, and from what is stated in the caption of Fig 2, these are composed of DMPC:DMPG:DHPC = 8:2:20. This description, lacking details on lipid:protein ratio, concentrations, or buffer in which bicelles are formed, is totally insufficient to allow reproduction of the experiments. Note that Nath et al (2014) and Hervas et al (2017) already saw that lipid composition and membrane curvature greatly influenced Atg3 activity, thus the authors must justify why they have chosen this composition and this system.

Author reply: We apologize for not including these details in our first submission and have provided them in the Fig. 2 legends. In addition, details of NMR sample preparations are supplied in the NMR Spectroscopy section in Methods.

We have previously demonstrated that bicelles (DMPC:DMPG:LPE:DHPC =5:2:3:40, q=0.25, and DMPC:DMPG:LPE:DHPC=5:2:3:12.5, q=0.8) can replace liposomes in an *in vitro* conjugation assay (Ye et al 2021, Nat. Commun., supplementary Figs. 12 and 13). The ability of these bicelles to support hAtg3 activity is presumably because the loosely packed, dynamic planar surfaces of bicelles mimic a membrane with the type of packing defects that are required for interactions with these molecules. Please note that lyso-PE (14:0) was used as a reaction substrate since PE lipids are incompatible with bicelles. Additionally,

TROSY spectra of hAtg3 in bicelles with and without LPE show minor resonance differences (Extended Data Fig. 6a). These differences are ascribed to the general perturbations of bicelles by aliphatic chain of lysolipids since the spectra of hAtg3 in bicelles with LPE or LPC are nearly identical (Extended Data Fig. 6b). Therefore, we chose to use bicelles composed of DMPC:DMPG:DHPC =8:2:20 (q=0.5) in this study.

3. Again on the matter of lipid composition, there appears to be no PE in the bicelles used. Drawing conclusions from the behavior of ATG3 when binding to membranes in the absence of its main substrate PE, when it has been seen by several authors that, depending on the % PE, Atg3 activity can vary greatly, certainly limits the validity of the conclusions obtained by the authors. This point should be carefully discussed.

Author reply: We thank the reviewer for this important and insightful remark. We have included additional data (Extended Data Fig. 6) that indicate that bicelles with and without LPE interact with hAtg3 similarly. Please see our response above to Reviewer #1, comment (2).

4. Checking the effect of the different mutations in vivo using Atg3-KO MEF provides very clear and consistent results with what was obtained in vitro. However, in addition to measuring basal autophagy by treating the cells with bafilomycin, it could have been tested how they behave in autophagy caused by other agents, e.g. starvation, treating the cells with nutrient-poor medium, or with rapamycin.

Author reply: We appreciate the suggestion to explore the behavior of the Atg3 mutants in induced autophagy. As recommended, we performed additional experiments to evaluate the effects of F296 mutants on nutrient starvation-induced LC3-II synthesis and lysosomal turnover, providing insights into autophagic flux in MEF cells. Our new *in vivo* data, consistent with *in vitro* results, demonstrated clear and consistent outcomes. Moreover, we assessed the lysosomal turnover of LC3-II using the lysosomal inhibitor bafilomycin A1. The data from these experiments (new Fig. 4) provide extensive evidence of the effects of the Atg3 mutants under different autophagic conditions. We thank you for your valuable suggestion, as it has enhanced the significance and robustness of our study.

5. In the discussion section, one misses the role of the F and G helices when Atg3 interacts with the other components of the conjugation system, such as ATG7, ATG8 or the E3 complex. This would provide the reader with an idea on how the complete system works, and the importance of the F and G helices in Atg3 operation.

Author reply: We thank the reviewer for this excellent suggestion. We have added a paragraph to discuss the implications of these newly uncovered membrane interactions of the C-terminal in the context of the LC3-Atg3 intermediate and the E3 complex. Our analysis of an existing Atg8-yAtg3 structural model and preliminary NMR data on LC3B-hAtg3 (conjugated through a disulfide bond) in the absence and presence of bicelles support an assumption that the multifaceted membrane interaction of hAtg3 applies to the LC3B-hAtg3 intermediate.

Reviewer #2 (Remarks to the Author):

The manuscript by Ye et al. (NCOMMS-23-03873, entitled “Translating Membrane Geometry into Protein Function: Multifaceted Membrane Interactions of Human Atg3 Promote LC3-Phosphatidylethanolamine Conjugation during Autophagy”) shed light on a very important and still not completely understood aspect of autophagosome biogenesis – molecular mechanism of Atg8 lipidation. Using NMR spectroscopy they were able to solve the solution structure of human ATG3 protein (its essential core compartments except the N-terminal amphiphilic α -helix) and demonstrate the conformational plasticity of the ATG3 C-terminal locus

around catalytic Cys264. They also showed by rationally-designed mutagenesis that flexibility of the C-terminal region is dependent on the protonation state of the invariant His266, which appeared as a unique pH sensor in human ATG3. His266 protonation stabilizes inactive conformation of the ATG3, while deprotonation of His266 – in higher pH values or by H266L mutation – stabilizes active ATG3 conformation and switch on the catalytic E2 activity of the enzyme. Most interesting finding within the presented work is, in my opinion, that this flexible C-terminal part of the ATG3 is also interacting with membranes, organizing a close proximity of ATG3 catalytic Cys264, its substrate ATG8 and the second substrate, phosphatidylethanolamine (PE) near the highly curved phagophoric rim for effective ATG8-PE conjugation.

The manuscript is well written and clearly organized, cited literature is enough and sufficient to introduce readers into the subject. All experiments (NMR for structure determination and for evaluation of relative mobility of ATG3 regions, in-vitro reconstitution of Atg8 lipidation, and mutational analysis of the desired ATG3 residues) are at the high standards. The resulting conclusions are of sufficient novelty and of considerable biological interest, and in the discussion the authors convincingly point out that their work opens up several ways for follow-up studies geared at understanding the precise mechanisms of Atg8 lipidation. In this way, I found that the manuscript is suitable for publication in Nature Communication after some minor revisions.

Author reply: We highly appreciate the feedback and encouraging remarks.

Major point:

The obvious drawback of this study is that the obtained results and conclusions are based mostly on in-vitro experiments, limiting the understanding of real role of the discovered ATG3 C-terminal membrane-binding site on Atg8 lipidation. Connected to that, autophagy-related E3 complex (ATG12~ATG5/ATG16L1) is essential for the effective in-vivo lipidation of ATG8-proteins. The authors performed in-vitro LC3 lipidation assays without E3, using ability of the Atg8 to conjugate lipids in presence of Atg7 (E1), Atg3 (E2), Atg8 and lipids (liposomes), reported from very first works on ATG3 enzymatic activity. The authors should at least discuss the relation between in-vitro and in-vivo catalytic activity of ATG3 and extrapolate conclusions on the complete Atg8-PE enzymatic cascade.

Author reply: Thank you for an excellent suggestion. We have added a new paragraph in the Discussion section to discuss the implications of these newly uncovered membrane interactions of the C-terminal in the context of the LC3-Atg3 intermediate and the E3 complex. Our analysis of an existing Atg8-yAtg3 structural model and preliminary NMR data on LC3B-hAtg3 (conjugated through a disulfide bond) in the absence and presence of bicelles support an assumption that the multifaceted membrane interaction of hAtg3 applies to the LC3B-hAtg3 intermediate.

Minor points

1. Lines 49-50 – “lipidation ... to” is not correct statement, better to say “lipidation by” or “conjugation to”

Author reply: Corrected.

2. Line 70 – “concurrently”? more precise description will be “sequentially”, as the later experiments shows that C-terminal ATG3 region interacts with the membrane only after the NAH is bound.

Author reply: Corrected, and thank you for the insight.

3. Line 74 – “substrates of LC3 and PE lipids” should be “substrates – LC3 and PE lipids”

Author reply: Corrected.

4. Lines 113 and 127 – *The deuterium exchange is not the method by which one can measure real stability, but rather flexibility of particular protein regions. I would recommend to avoid the term stability and state what helix F is more mobile rather than “less stable”*

Author reply: We agree with the reviewer's remark and have corrected the text accordingly.

5. Line 138 – *“colocalizes” is not the proper term here, better to state “is in close proximity”*

Author reply: Corrected.

6. Lines 163-164 and Extended Data Fig. 4b – *seems, authors assumed “the CSP induced by bicelles”.*

Author reply: Corrected.

7. Line 202 – *better “Analysis by mutagenesis” or “Mutational analysis”*

Author reply: Corrected.

8. Lines 207-208 – *provide reference on the used method to predict secondary structure by Ca/Cb resonances values.*

Author reply: We have provided references on using $^{13}\text{C}_\alpha$ and $^{13}\text{C}_\beta$ secondary chemical shifts to predict secondary structures in the Extended Data Fig. 7 legend.

9. Line 337 – *inability of ATG3 C-terminal region to interact with membrane without membrane association by N-terminal helix is one of the key result in the manuscript. I would additionally mention this in the Results section.*

Author reply: Thank you for this excellent suggestion, and we have moved this observation to the Results section.

10. Lines 373-374 – *difference between yeast Atg3 and human enzymatic mechanisms are supported by the fact that yAtg3 has an active AIM/LIR motif, while higher organisms evolved to lose the LIR from Atg3.*

Author reply: Thank you for this great suggestion, and we have included it in the Discussion section.

11. Figures 3a, 3c and Extended Data Fig. 2 – *will be better to have human protein sequences at the first raw.*

Author reply: Updated.

12. Figures 3b, 3d and 4a – *indicate which time was used in CL (control probes w/o liposomes) in the figure legends.*

Author reply: The incubation time used in CL was 180 mins. We have provided this experimental detail in

figure legends (Fig. 3 and Extended Data Fig. 3b).

Reviewer #3 (Remarks to the Author):

The authors of this study correlated structural characteristics of ATG3, revealed by NMR, with its function in autophagy. For NMR studies, a truncated variant of ATG3 lacking its N-terminus, which contains an amphipathic membrane binding helix, and a central part including aa 90-190, was used. This variant was designed based on the crystal structures of yeast Atg3 and Atg3 from *A. thaliana*, in both of which this region was not visible in the electron density. Not surprisingly, the solution structure of hATG3 superimposed very well with the crystal structures of yeast and plant Atg3. Importantly, the deleted region contained the ATG7 binding site, resulting in a strongly impaired catalytic activity of ATG3 delta90-190 to conjugate LC3 to PE. The authors reported a similar structure in their Nat. comm. paper in 2021. In this study, they expanded their characterization of ATG3 delta90-190 and determined exposed aa based on H/D exchange as well as membrane binding using bicelles. They found that the catalytic loop of ATG3 is highly dynamic and in close proximity with bicelles. In their previous publication, they already reported chemical shift perturbations near the catalytic site upon incubation of ATG3 delta90-190 with bicelles, but 44 residues remained unassigned. In this study, they optimized ATG3 delta90-190 by introducing four amino acid substitutions which stabilized ATG3 on bicelles and improved the NMR signals so that most aa could be assigned. Interestingly, this variant showed stronger chemical shift perturbations (CSPs) of residues in the C terminus of ATG3 compared to the previously reported CSPs of ATG3 delta90-190. The authors continued by performing cross-saturation experiments, revealing that besides residues of the N terminal amphipathic helix, residues that cluster around the catalytic site are in close proximity to lipids of bicelles. Secondary chemical shift analysis implied that residues 265-277 form a helix upon membrane binding with amphipathic character. However, as opposed to the N terminal amphipathic helix, this helix does not insert into the lipid bilayer. Mutating hydrophobic residues of that helix to aspartate interfered with LC3B lipidation in vitro. Residues of the second membrane binding site formed a largely hydrophobic helix and mutation of these residues either destabilized the protein or impacted on its catalytic activity. Notably, H266 was found to be pH sensitive and the interaction of ATG3 with membranes as well as its catalytic activity were abolished if H266 was protonated or mutated to Lysin, while mutation to leucine or phenylalanine rendered ATG3 pH insensitive, being active also under acidic conditions. However, the H266L or H266F mutants failed to complement ATG3 in ATG3KO MEFs, while H266K partially restored lipidation, which is in odds with observations made in vitro.

Overall, this study provides only incremental advances over to the previous study of the authors published in Nat. comm. in 2021. They expanded on the previous observation that residues near the catalytic site bind membranes, which would be expected because finally LC3 needs to be transferred to PE, which requires both entities to be in close proximity. Data that go beyond the previous study include identification of an amphipathic helix that does not insert into the membrane but contains a pH sensitive His that promotes LC3 lipidation only in its unprotonated state. What role this pH sensitivity plays during autophagy remains uncertain because the pH of the cytosol is with 7.4 in between the pH optimum of ATG3 (pH = 8.5) and the pH at which the activity of ATG3 is abolished (pH = 6). Moreover, there is a contradiction between the activity of ATG3 H266 mutants in vitro and in vivo, making it difficult to assign a specific function to H266 during LC3 lipidation.

Moreover, the study was conducted in the absence of LC3, which might has a profound impact on the interaction of the two helices F and G of ATG3 with membranes. Overall, the new data provided by this study does not merit publication in Nat. commun. and the study appears to be too preliminary for publication in a more specialized journal. Notably, the contradiction of results from in vitro and in vivo

experiments needs to be resolved and LC3 needs to be taken into account to reveal meaningful information on the orientation of the LC3-ATG3 conjugate on membranes. Finally, ATG12-5-16L1 also impacts on the recruitment and, presumably, the orientation of LC3-ATG3, arguing that the mechanism of ATG3 action can only be revealed if ATG3 is studied in the context of the E3-ligase and LC3.

Author reply: Thank you for your extensive and important feedback.

As noted by Reviewers #1 and #2, this study's most significant result and innovation are that the catalytically important C-terminal parts of hAtg3 directly interact with membranes (in addition to its well-known N-terminal membrane curvature-sensitive helix). This result has several profound implications. First, the multifaceted membrane interaction provides a molecular basis that directs the catalytic residue Cys264 of hAtg3 to the membrane surface and brings its substrates (LC3 and PE lipids) into proximity (this has been a long-standing question for Atg3 action). Second, the multifaceted membrane interaction of hAtg3 functionally links its conjugase activity to curvature sensitivity, thereby advancing the concept that the distinct structure of a growing phagophore directly regulates autophagosome biogenesis. Third, the membrane curve-selective interaction is a recurring theme for autophagy-related proteins, including Atg1, Atg3, Atg13, Atg14L, and Atg16L. In fact, the number of proteins reported to depend on membrane geometry for function has steadily increased over the last 15 years. Yet the structural and molecular mechanisms of their actions are largely unknown. This study of hAtg3 is the first example to show how membrane structure can be translated into protein function by a multifaceted membrane interaction mechanism. We agree with this reviewer that further studies of the multifaceted membrane interaction in the context of the E3-ligase and LC3 are needed to understand the mechanism of Atg3 action fully. We are working towards this ultimate goal despite inherent challenges associated with the increased size of a large complex, and those studies are beyond the scope of this work. However, as described in a new paragraph in the Discussion section, NMR preliminary data on LC3B-hAtg3 and analysis of an existing Atg3-yAtg8 structural model support the notion that the newly uncovered interactions of the C-terminal with the membrane are relevant in the context of LC3B-hAtg3.

We agree with this reviewer that it is unclear whether the pH-dependent conjugase activity of Atg3 plays any role during autophagy. Therefore, we have removed results related to the role of H266 in hAtg3's *in vitro* pH sensitivity. The *in vitro* and *in vivo* activities of hAtg3^{H266K} mutant reasonably agree (nearly abolished activity in the absence of E3 enzyme *in vitro* vs. markedly reduced activity in the presence of E3 enzyme *in vivo*), but hAtg3^{H266L} does not. These results suggest that His266 may serve additional functions *in vivo*, such as product release or substrate reloading, but further study is needed, as this reviewer noted.

In addition, we would like to clarify several issues raised by the reviewer.

(1) The structure of hAtg3^{ΔN25} described in our previous study is a computational model using the Chemical-Shift-ROSETTA (CS-ROSETTA) software (Ye et al 2021, Nat. Commun.). In contrast, the hAtg3^{ΔN25, Δ90-190} structure reported in this study is the first experimentally determined structure for human Atg3 (PDB accession ID 8FKM). While its core structure is similar to the structures of yAtg3 and AtAtg3, we observed substantial differences in the structure and property hAtg3 helix F (Fig. 1b). This helix and the catalytic loop are conformationally flexible based on NMR H/D exchange experiment. We suggest that the flexibility is important for structural rearrangements that are critical for hAtg3 conjugase activity.

(2) The hAtg3^{Δ90-190} construct retains about 50% of its conjugase activity compared to the wildtype protein *in vitro*, and its reduced activity is at least partially due to a slow formation of the LC3B- hAtg3^{Δ90-190} intermediate as a result of removing its primary Atg7 interacting site (Ye et al 2021, Nat. Commun., supplementary Fig. 8). In this study, we introduced four mutations to this construct to stabilize its interaction

with bicelles. The new construct (hAtg3^{Δ90-190, 4M}) remains functional *in vitro*. It produces dramatically improved spectra in bicelles (Extended Data Fig. 3). This allowed us to achieve near-complete backbone resonance assignments, obtain a more complete profile of chemical shift perturbations (CSPs) induced by bicelles, and directly map the protein/lipid interaction interface using an NMR cross-saturation experiment.

(3) Unlike chemical shift perturbations that are sensitive to both direct interactions and conformational changes (allosteric effects), cross-saturation experiments provided tangible experimental evidence that, in addition to its N-terminal membrane curvature-sensitive helix, the C-terminal catalytic important regions of hAtg3 directly interact with the membrane. In conjunction with NMR paramagnetic relaxation enhancement experiments, we conclude that the N-terminal helix of hAtg3 inserts into the membrane and interacts with the hydrophobic core, and the C-terminal catalytic important regions interact with the polar headgroup layer of the membrane.

REVIEWERS' COMMENTS

Reviewer #1 (Remarks to the Author):

The authors have replied satisfactorily to all my criticisms.

Response to the Reviewers

Reviewer #1 (Remarks to the Author):

The authors have replied satisfactorily to all my criticisms.

Author reply: We sincerely appreciate all reviewers for their constructive and supportive comments.